# Disease-Modifying Effects of Non-Invasive Electroceuticals on β-Amyloid Plaques and Tau Tangles for Alzheimer’s Disease

**DOI:** 10.3390/ijms24010679

**Published:** 2022-12-30

**Authors:** Junsoo Bok, Juchan Ha, Bum Ju Ahn, Yongwoo Jang

**Affiliations:** 1Department of Medical and Digital Engineering, College of Engineering, Hanyang University, Seoul 04736, Republic of Korea; 2Department of Biomedical Engineering, College of Engineering, Hanyang University, Seoul 04736, Republic of Korea; 3Department of Pharmacology, College of Medicine, Hanyang University, Seoul 04736, Republic of Korea

**Keywords:** Alzheimer’s disease, β-amyloid plaque, tau tangle, electroceuticals, electroceutical therapy, electrical stimulation

## Abstract

Electroceuticals refer to various forms of electronic neurostimulators used for therapy. Interdisciplinary advances in medical engineering and science have led to the development of the electroceutical approach, which involves therapeutic agents that specifically target neural circuits, to realize precision therapy for Alzheimer’s disease (AD). To date, extensive studies have attempted to elucidate the disease-modifying effects of electroceuticals on areas in the brain of a patient with AD by the use of various physical stimuli, including electric, magnetic, and electromagnetic waves as well as ultrasound. Herein, we review non-invasive stimulatory systems and their effects on β-amyloid plaques and tau tangles, which are pathological molecular markers of AD. Therefore, this review will aid in better understanding the recent technological developments, applicable methods, and therapeutic effects of electronic stimulatory systems, including transcranial direct current stimulation, 40-Hz gamma oscillations, transcranial magnetic stimulation, electromagnetic field stimulation, infrared light stimulation and ionizing radiation therapy, and focused ultrasound for AD.

## 1. Introduction 

As the most common type of dementia, Alzheimer’s disease (AD) is a progressive neurodegenerative disease that gradually deteriorates cognitive functions such as thinking, remembering, and reasoning over several years [1,2,3]. The major pathological phenotype of AD is brain shrinkage (atrophy), which generally worsens as the disease progresses [4]. In the early stages of AD, neuronal degeneration is clearly observed in the hippocampus, which is a critical component of the limbic lobe for learning and memory. Through magnetic resonance imaging (MRI), image analysis of the atrophy of the hippocampal region can be used for identifying structural changes in the diagnosis of AD [5].

Although the pathological cause of AD has not been clearly identified, extracellular amyloid-β (Aβ) plaques and intracellular neurofibrillary tau tangles have been observed in the brains of patients with AD. Numerous studies have shown that Aβ plaques and/or tau tangles in the brain are important molecular markers that initiate a deleterious cascade triggering neurodegeneration in AD [6,7,8]. Therefore, the removal (clearance) of Aβ plaques and tau tangles is considered to be critical for treating AD [9,10]. Hence, researchers have attempted to develop a therapeutic drug to attenuate the levels of Aβ plaques and/or tau tangles in the brain. However, developing therapeutic agents, such as small molecules and antibodies, for Aβ and tau proteins through the blood–brain barrier (BBB) is difficult. 

In this regard, non-invasive brain stimulation using electrical systems appears to be a promising approach for alleviating the pathological symptoms in the brains of patients with AD [11]. In the brain, neurons are electrically excitable cells that can be activated through a change in membrane voltage in response to various stimuli. Some types of glial cells, such as astrocytes, also exhibit excitable attributes that regulate neighboring neurons. Therefore, the electrical stimulation of brain tissue can influence brain activity. To date, extensive studies have been conducted to elucidate the disease-modifying effects of AD through the electric stimulation of brain circuits using various advancing technologies. 

Recently, electroceuticals have been defined as a new category of therapeutic agents that function by targeting neuronal circuits in the brain [12]. The scope of electroceuticals is gradually expanding from electrical stimulation to various stimulations generated by electric systems, such as magnetic fields and ultrasonic waves. Depending on the stimulation type and target disease, electroceuticals can be divided into implantable and wearable systems [13,14]. For applications and therapeutic effects, extensive studies have been conducted to achieve device miniaturization, biocompatibility, biodegradability (a certain case), and functionality based on interdisciplinary biomedical engineering and medical science. Several studies have attempted to optimize the variable conditions of electrostimulation, such as the electrode position, intensity, frequency, duration, and treatment sessions, to maximize the therapeutic effect. Therefore, we review non-invasive stimulatory systems and their effects on Aβ plaques and tau tangles in AD based on the research fields of electroceuticals. To achieve this objective, we categorize electrical stimulation based on energy forms such as electric, magnetic, and electromagnetic waves and ultrasound (Figure 1).

## 2. Electrical Fields

In the electroceutical field, electrical stimulation is a representative stimulatory method that applies an electric current through electrodes placed in specific regions. Generally, electrical stimulation is implemented using various waveforms (Figure 1A) by directly applying a current to the target area inside or outside the human body. For electrical stimulation, various regulatory factors, such as the intensity, cycle, and duration of the stimulus, are critical in determining the therapeutic effect and side effects. Therefore, optimizing these stimulatory conditions for a targeted disease is important for obtaining the desired effect. The preventive and/or therapeutic effects of electrical stimulation have been actively investigated for AD. Electrical stimulation affects the degradation of Aβ plaques and tau tangles by regulating the immune system. 

### 2.1. Transcranial Direct Current Stimulation (tDCS)

tDCS is a neuromodulatory technique that supplies a low electric current to the scalp of the brain. The electric field is generated between an active electrode and a counter electrode attached to the scalp, which can directly apply a low electric current (<2 mA) to a specific area of the brain. Neurostimulation with a low electric current induces changes in the membrane potential of neighboring cells around a stimulatory region, ultimately resulting in changes in the neuroactivity and excitability of the cerebral cortex [15]. In addition, tDCS regulates the activation of glial cells involved in the clearance of Aβ plaques and tau tangles in the brain and suppresses inflammatory mediators such as nuclear factor κB (NF-κB) and tumor necrosis factor alpha (TNF-α) (Figure 2A).

Electrical stimulation site(s) attached to electrodes may vary depending on the target disease, based on an electrode-positioning criterion in an electroencephalogram (EEG), such as the international 10–20 electrode system, neuro-navigation system, and physiology-based placement [16]. The tDCS technique has primarily been studied to provide relief from migraine or from symptoms of cognitive dysfunction in patients with AD, mild cognitive impairment (MCI), and attention deficit hyperactivity disorder (ADHD). Thus, the electrode position for this neuro-stimulation primarily targets the frontal cortex or dorsolateral prefrontal cortex, which is involved in the control of cognitive functions, as long as the stimulation intensity is limited to 2 mA or less. In addition, the effectiveness verification of cognitive function is primarily performed using the change in EEG measurements and cognitive tasks for diagnosing cognitive impairment, such as the Mini-Mental State Examination and the Trail Making Test. With the development of AD-related animal models, through verification of the effectiveness of tDCS on cognitive dysfunction, researchers are attempting to interpret the action mechanism of tDCS at the molecular and cellular levels using experimental techniques, such as tissue staining, immunohistochemistry, Western blotting, and real-time polymerase chain reaction (RT-PCR). In particular, numerous studies have focused on the changes in tau tangles and Aβ plaques after tDCS treatment. 

#### 2.1.1. Tau Tangle

Over the past decade, the effects and mechanisms of tDCS on tau tangles have been extensively studied. In 2015, Yu et al. investigated the effect of tDCS in Sprague Dawley (SD) rats (8 weeks, 250–320 g) injected with 5 µL of synthetic amyloid beta peptide 1–40 (Aβ40) into the CA1 region of the hippocampus [17]. For the active electrode, a modified plastic tube electrode was installed above the right frontal cortex. The counter electrode was installed in a corset in contact with the ventral thorax. tDCS stimulation was performed in 10 sessions for 2 weeks, with each session at anodal current intensities of 20, 60, 100, and 200 µA for 20 min, and a sham control was stimulated at 100 µA for 10 s. Consequently, no significant difference was observed in the neurofibrillary tangle (NFT)-like changes caused by tau inside the neuron fibers, dendrites, and axons between two groups with Aβ40 injection in the presence and absence of tDCS stimulation. Nevertheless, the application of 100 or 200 µA tDCS improved the spatial learning and memory impairments observed in Aβ-treated rats. The authors also observed that the expression of inflammatory factors, such as NF-κB and TNF-α, which are related to the decline of spatial memory and learning, decreased in the 100 and 200 µA tDCS-treated groups. These findings prove that tDCS stimulation can suppress Aβ40-induced inflammatory mediators such as NF-κB and TNF-α, preventing the decline in learning and memory. This anti-inflammatory effect may be caused by the anodal stimulation that continuously depolarizes the membrane potential of astrocytes, thereby reducing their vulnerability to Aβ [18].

In contrast, a study by Gondard et al. in 2019 demonstrated a decrease in tau tangles after tDCS treatment in 3xTg AD mice [19]. For electrical stimulation, two paddle electrodes were implanted on the surface of the skull bone. The cathodal electrode was positioned 2 mm anterior to the bregma perpendicular to the interfrontal suture, and the other anodal electrode was placed 1 mm anterior to the lambda above the dorsal temporal hippocampus, including the CA1 region and dentate gyrus. tDCS stimulation was conducted in 15 sessions for 3 weeks, and a current of 50 mA was maintained for 20 min in each session. Thereafter, the authors compared the protein levels of total tau, Ser416 residue-phosphorylated tau (pTau), and amyloid precursor protein (APP) in the hippocampus and cortex. Although a decreased expression of tau and pTau was observed in the tDCS-treated hippocampus, the decrease was statistically insignificant [19].

#### 2.1.2. β-Amyloid Plaques

Various studies have demonstrated the therapeutic effects of tDCS on Aβ plaques compared with that on tau tangles. In particular, Luo et al. recently verified the effects of tDCS in three different studies using APP/presenilin-1 (PS1) transgenic mice. Their first study in 2020 investigated the effect of tDCS on spatial learning/memory and attenuation of the Aβ42 peptide burden [15]. Subsequently, they examined its effect on cognitive impairment in 2021 [20]. Most recently, in 2022, they attempted to determine its effect on Aβ production [21].

In subsequent experiments, the anodal electrode was implanted over the frontal cortex. The cathode was attached to the chest and abdomen. tDCS stimulation was performed for 10 sessions in 2 weeks, and each session was conducted for 30 min with a current of 150 µA. As a control, a sham was stimulated electrically with the same position and stimulatory conditions for only 10 s. The experiments were classified into four groups: wild-type (CTL), APP/PS1 (AD), APP/PS1 with sham stimulation (ADST), and APP/PS1 with tDCS stimulation (ADT). First, using experimental analyses such as Western blotting, immunofluorescence staining, and optical density [15], the authors observed that the protein expression of Aβ42 decreased significantly in the hippocampus of the ADT group compared with the AD and ADST groups. In addition, tDCS-stimulated hippocampus exhibited a significant decrease in glial fibrillary acidic protein (GFAP), which is primarily expressed in astrocytes, and a higher expression of neurofilament 200 (NF200), which is a high-molecular-weight neurofilament protein. These changes in Aβ42, GFAP, and NF200 were observed in the dentate gyrus (DG), CA1, and CA2-3 regions, and the decrease in Aβ42 expression was the largest in the DG area. In a subsequent study, they observed that tDCS stimulation increases the protein level of NeuN, which is specifically located in neurons and decreases the levels of the apoptotic factor caspase 3 in the hippocampus [21]. Notably, further investigation revealed a molecular mechanism for reducing the expression level of Aβ in the frontal cortex and hippocampus following exposure to tDCS. Repetitive tDCS stimulation suppressed the production of Aβ by decreasing the levels of APP and beta-secretase 1 (BACE1), which are important for the generation of Aβ in neurons, and by increasing the level of a disintegrin and metalloproteinase domain-containing protein 10 (ADAM10), which modulates dendritic spine formation [22]. Overall, tDCS stimulation of the hippocampus is likely to suppress the production of Aβ and promote the Aβ clearance system in astrocytes [23,24]. In addition, stimulation seems to increase neuronal survival by exerting a neuroprotective effect through the regulation of the apoptotic pathway and downregulation of Aβ production. Additionally, a recent study by Duan et al. compared the effects of tDCS on Aβ between anodal and cathodal stimulation in APP/PS1 mice [16]. The active electrode was installed over the left side of the prefrontal cortex, and the counter electrode was attached to the central thorax. A current of 300 µA was applied to the active electrode, and stimulation was performed for 20 min every day for five consecutive days. As a control, the sham group received the same electrodes without electrical stimulation. They compared the levels of Aβ between anodal and cathodal stimulation groups. However, in both groups, no significant effect on the total Aβ concentrations in the hippocampus was observed. When the current density applied was calculated, it was significantly lower than that of other studies that indicated a significant Aβ reduction. This finding suggests that the current density is an important factor that determines the efficacy of tDCS in AD. All results regarding tDCS are summarized in Table 1.

Several recent studies have investigated the effects of tDCS on the pathological symptoms of AD from the cognitive behavioral level to the molecular level [25,26,27,28,29]. Despite recent advances in tDCS stimulation using electroceuticals, the disease-modifying effects of tDCS on AD remain controversial. Thus, optimizing and standardizing the stimulatory conditions and electrode positions of tDCS for the treatment of AD is challenging.

### 2.2. 40-Hz Gamma Oscillations Using Electrical Stimulation

In the brain, spontaneous electrical activity that is derived from neurons represents the macroscopic activity of a certain area of the brain over a period, which can be recorded using EEG as an electrogram. Transient rhythmic activity is generally divided into bands of frequency, such as delta (<4 Hz), theta (4–8 Hz), alpha (8–13 Hz), beta (13–30 Hz), and gamma (>30 Hz) [30]. Among these frequency bands, the disruption of gamma oscillations between 20 and 50 Hz is linked to various neurological disorders [31,32]. Iaccarino et al. observed a reduced gamma power (20–50 Hz) during hippocampal sharp-wave ripples in 5xFAD mice. They demonstrated that gamma frequency entrainment in the hippocampus reduced the level of Aβ plaques through the activation of microglia [33]. Thereafter, an increasing number of studies have investigated the effects of hippocampal 40-Hz oscillations on AD. In most animal studies, hippocampal stimulation to induce 40-Hz oscillations has been performed using direct brain stimulation, optogenetic stimulation, and gamma entrainment using sensory stimulation [34]. Most of these studies have demonstrated that 40-Hz gamma stimulation modulates AD-specific pathology or improves behavioral and cognitive memory performance [35,36].

Furthermore, the effect of gamma entrainment using audiovisual stimulation was investigated in patients with MCI. In this study, 10 patients were stimulated for 4 or 8 weeks with a stimulator composed of goggles and headphones for light and sound stimuli, respectively [37]. As a result, the functional connectivity between the posterior cingulate cortex and precuneus in EEG measurements increased. In addition, the study observed reduced levels of immunity factors such as transforming growth factor alpha, macrophage inflammatory protein 1β, delta/notch-like epidermal growth factor receptor, and interleukin-5 in the cerebrospinal fluid (CSF) after 8 weeks. However, no significant change was observed in Aβ and tau levels in the CSF. 

For practical application in patients with AD, a non-invasive system that can provide electrical stimulation with the desired amplitude and frequency for gamma frequency entrainment is required. Transcranial alternating current stimulation (tACS) is a non-invasive stimulation method that applies a sinusoidal electric current directly with low intensity to the brain through electrodes placed on the scalp, which results in a change in endogenous cortical oscillatory rhythms. Recent studies have demonstrated that this transcranial stimulation can drive a 40-Hz stimulation to the hippocampus in diverse ways, thereby verifying the disease-modifying effects on AD.

Most studies have observed the amelioration of impaired cognition and memory in AD-related models and patients with AD [36]. First, invasive methods have been applied, such as deep brain and optogenetic stimulations for 40-Hz synchronization. For instance, chronic deep brain stimulation has been demonstrated to improve synaptic plasticity and neurogenesis, thereby modulating local neural activity, and stimulating the progressive reorganization of neural circuits [38,39]. In a stimulation method using optogenetic tools, the gamma power of the basolateral amygdala increased during contextual consolidation, and gamma synchronization was observed to improve the subsequent memory strength [40]. Moreover, a 40-Hz optogenetic stimulation of medial septal parvalbumin-positive neurons restored spatial impaired memory by restoring hippocampal gamma oscillations and theta-gamma phase–amplitude coupling [41].

A recent investigation performed in 2022 by Dhaynaut et al. demonstrated the tau burden-removal effect of tACS in four AD patients with Aβ plaques [42]. Two electrodes were attached to the bilateral temporal lobes by applying a sinusoidal wave of 40 Hz with an amplitude of up to 2 mA. The tACS treatment was conducted for 1 h every day, five times a week, for a total of 20 sessions [42]. Using positron emission tomography (PET) imaging, they observed a significant decrease in the level of pTau but did not observe any significant difference in the levels of Aβ and microglial activation. However, the level of Aβ was notably reduced in one AD patient treated with tACS (−5.4%).

Vagus nerve stimulation (VNS) is another 40-Hz stimulation technique that modulates the vagus nerve, which originates from the brain, with electric impulses. The vagus nerve travels between the peripheral organs and lower part of the brain through the neck. Thus, non-invasive VNS to the left cervical or auricular transcutaneous vagus nerve can transfer electric pulses to the brain via the vagus nerve. In particular, VNS has been used for symptom relief in patients with various neurological diseases for whom therapeutic drugs and incisional surgery are impossible [43].

To verify the effects of VNS, numerous studies have investigated the therapeutic effects of cognitive and memory impairments in AD-related animal models and AD patients. This effect is considered to be due to the structure and function of the vagus nerve, which has an extensive innervation to visceral organs, including the auricular vagus nerve branch [11]. Anatomically, the vagus nerve consists of several main nuclei, including the nucleus of the solitary tract (NTS) and locus coeruleus (LC) of the brain (Figure 2B). Because of innervated projections to central nervous neurons, the vagus nerve is believed to be involved in hippocampal stimulation to modulate cognitive impairments [43]. Recent studies have been conducted on the activation of the LC-norepinephrine system for AD treatment [11,44]. Remarkably, Yu et al. demonstrated the possibility of AD treatment through transauricular VNS (taVNS) using a 40-Hz oscillation suitable for AD treatment instead of the 20–30 Hz used in previous studies [45,46,47]. When taVNS was applied at a frequency of 40 Hz to APP/PS1 mice, significant decreases in Aβ42 expression and soluble Aβ40 and Aβ42 levels were observed in the hippocampus. Moreover, a 40-Hz stimulation of taVNS was observed to induce microglial phagocytosis and regulate microglial pyroptosis through the inhibition of the P2X7R/NLRP3/caspase-1 pathway in the hippocampus [48]. Moreover, a 40-Hz stimulation of taVNS has a neuroprotective effect on hippocampal neurons through the suppression of inflammation by inhibiting the NF-κB pathway [48]. All results regarding 40-Hz oscillations are summarized in Table 1.

Numerous studies have demonstrated the therapeutic effects of 40-Hz oscillations in AD. As non-invasive methods, tACS and VNS are likely to be useful stimulatory systems for transferring 40-Hz oscillations to the brain (Figure 2). Recently, 40-Hz oscillations using tACS and taVNS have shown the therapeutic effects of Aβ plaques and tau tangles in the hippocampus. Further investigations are required in various AD-related animal models and patients with AD.

**Table 1 ijms-24-00679-t001:** A summary of electric stimulation on Aβ and tau pathology.

Stimulation	Stimulation Intensity	Duration	Subject	Main Finding	Reference
tDCS stimulation					
Active electrode: right forntal cortex Counter: vental thorax	+0.2 mA	20 min 10 sessions in 2 weeks	Female Sprague Dawley rats Aβ injected in hippocampus	- ↑ density of nissl’s body in deeper hippocampus - ↑ ChAT level - no significant difference in the neurofibrillary tangles (NFT)-like changes - ↓ spatial learning and memory dysfunction	[17]
Active electrode: anterior to lambda Counter: anterior to bregma	+0.05 mA	20 min 15 sessions in 3 weeks	3xTg AD	- Insignificant change in total tau, pTau and APP level - no effect in improving memory performance	[19]
Active electrode: frontal cortex Counter: chest and abdomen	+0.15 mA	30 min 10 sessions in 2 weeks	APP/PS1 C57 mouse	- ↑ NF200 level in hippocampus - ↓ Aβ42 level in hippocampus - ↓ GFAP level in hippocampus - ↑ spatial learning and memory in the early stage APP/PS1 transgenic mouse	[15]
				- ↓ Aβ42 level in hippocampus - ↓ GFAP level in hippocampus - ↑ spatial learning memory and recognition memory	[20]
				- ↓ Aβ level in hippocampus and frontal cortex - ↓ level of Iba1 and GFAP in hippocampus and frontal cortex - ↑ ADAM 10, NeuN, LRP1 and PDGRFβ hippocampus and frontal cortex	[21]
Active electrode: left prefrontal cortex Counter: ventral thorax	+0.3 mA	20 min 5 sessions in 5 days	APP/PS1 B6C3 mouse	- No significant effect on total Aβ concentrations in hippocampus - ↑ spatial learning, recognition and working memory	[16]
	−0.3 mA				
40 Hz gamma oscillation					
Flicking light and sound	-	1 h 4 or 8 weeks	10 MCI patients	- No significant changes in CSF Aβ and tau - ↑ functional connectivity between posterior cingulate cortex and precuneus - Downregulation of immune factors with engagement of the neuroimmune system - ↑ network functional connectivity after 8 weeks of daily flicker	[37]
tACS Electrode: bilateral temporal lobes	2 mA	1 h 4 weeks	4 AD patients	- ↓ significantly in pTau level in brain - No significant changes in level of Aβ and microglia activation - No significant changes in overall cognition	[42]
taVNS bilateral auricular concha	1.8 mA	30 min 15 days	14 APP/PS1 mice	- ↓ significantly in Aβ42 expression and soluble Aβ40 and Aβ42 levels in the hippocampus - ↓ P2X7R/NLRP3/caspase-1 pathway to regulate microglia pyroptosis - ↓ pro–IL-1β and pro–IL-18 to suppress inflammation - ↑ microglial phagocytosis - ↑ Spatial Learning and Memory	[47]

## 3. Transcranial Magnetic Stimulation (TMS)

A magnetic field is a vector field generated in the region around a magnetic material or moving electric charge. Generally, a magnetic field is induced through Faraday’s principle of electromagnetic induction when a high-voltage current passes through a coil (Figure 1B). When a magnetic coil is placed tangentially near the brain, the generated magnetic field penetrates the scalp and skull and then depolarizes neighboring neurons in the targeted brain areas. Based on this stimulation, TMS is a non-invasive method used to specifically stimulate a targeted region of the brain without requiring the attachment of an electrode to the brain surface. TMS stimulation does not require surgery, skin preparation, or an intravenous system [49]. 

Generally, TMS can be classified as single TMS (sTMS), paired-pulse TMS (ppTMS), and repetitive TMS (rTMS), based on the number of stimuli to the brain region. sTMS is used to analyze the motor cortical outputs and measure the central motor conduction time and motor-evoked potential [49]. Moreover, ppTMS has been applied to measure intracortical facilitation and motor cortical pathways [49,50]. rTMS delivers pulsed magnetic fields to modulate long-lasting cortical excitability [50,51,52]. rTMS has gradually gained interest in the treatment of various neurological disorders, including AD [49,53,54]. However, the mechanism(s) underlying its therapeutic effect remains unclear.

Several studies have reported the disease-modifying effects of TMS on the pathological symptoms of AD based on various stimulation parameters. Most studies observed the amelioration of the levels of Aβ burden and pTau tangles, which resulted in the rescue of impaired cognitive functions in various AD-related animal models. For instance, Wang et al. first demonstrated that TMS enhances the activity of the large conductance calcium-activated potassium (BK) channel by increasing the expression of homer1a, a scaffold protein in the hippocampus, which finally results in not only the reduction of Aβ plaques but also magnified hippocampal log-term potentiation (LTP) [55]. Using a magnetic stimulator, they applied chronic TMS (1, 10, 15 Hz, 250 µs duration, 80% of the maximum output of the machine for 5 s once a day) to the skulls of 3xTg mice for 4 weeks. As a result, TMS application at a frequency of 15 Hz recovered the suppressed activity of the BK channel by increasing the expression of homer1a, resulting in a lower Aβ burden in neocortical tissue (Figure 3) [55]. In addition, the administration of rTMS was observed to suppress the phosphorylation of tau and β-catenin proteins by influencing the expression of PS1, an upregulator of glycogen synthase kinase 3β (GSK-3β), which would promote the survival of neurons [56,57,58]. In Ref. [57], rTMS was performed with two frequencies (1 and 10 Hz), 30% of the maximum output (1.26 T), and two sessions of 1000 pulses for 14 days.

Remarkably, Ba et al. demonstrated a suppressed production of the Aβ peptide by inhibiting the transcription factor AP-1, finally resulting in reduced levels of APP in the CSF [59]. The molecular mechanism revealed that rTMS results in the reduction of APP through the inhibition of the MKK7-ERK1/2-c-FOS-APP axis in a 6-OHDA-induced mouse model of Parkinson’s disease (Figure 3) [59,60,61]. In addition, rTMS treatment restored impaired cognitive behaviors following intracranial injection of 6-OHDA in mice. For stimulation, rTMS was performed at two frequencies (1 and 10 Hz), 30% of the maximum output (1.26 T), and two sessions of 1000 pulses for 14 days. In another study, rTMS was applied to the parietal bone of mice using a magnetic-electric stimulator (CCY-3, Wuhan Yiruide Medical Equipment Co., Ltd., Wuhan, China) at 5 Hz, 120% of the average resting motor threshold, 600 pulses consisting of 20 bursts and 30 pulses each, and 2 s between trains for 14 days [62]. Consequently, this stimulation significantly reduced the expression of Aβ, APP, and pTau, which in turn not only recover the spatial learning and memory defects but alleviate the cognitive impairment of learning and memory in APP/PS1 mice [62]. Further mechanistic analysis demonstrated that rTMS treatment activates lysosomal degradation and clearance of Aβ plaques through decreased expression of apolipoprotein E (ApoE) and protein phosphatase 2A (PP2A) (Figure 3). Meanwhile, the increased level of the microtubule-associated protein 1 light chain 3 (LC3) II/LC3I ratio coupled with the decreased expression of p62 suggested that rTMS enhances hippocampal autophagy in APP/PS1 mice. Moreover, Lin et al. demonstrated that rTMS increases the drainage efficiency of the brain clearance pathway through the glymphatic system, parenchyma, and meningeal lymphatics, in addition to reducing Aβ deposits in 5xFAD mice [63]. Further immunohistochemical analysis revealed the promotion of neuronal activity and suppression of activated glial cells (microglia and astrocytes) in the prefrontal cortex and hippocampus of rTMS-treated 5xFAD mice [63]. rTMS stimulation was performed with a magnetic-electric stimulator (CCY-2, Wuhan Yiruide Medical Equipment Co., Ltd.) at 20 Hz, 1.38 T intensity, 2 min every two sessions consisting of 100 pulses in 40 trains, and 5 s between trains. Recently, Cao et al. demonstrated that rTMS activates phosphatidylinositol 3-kinase (PI3K) and protein kinase B (Akt) signals, which are involved in the cleavage of APP in SweAPP N2a cells and alleviation of cognitive deficits (Figure 3) [64]. The decrease in APP cleavage was attributed to a decrease in hippocampal Aβ levels. Using a magnetic stimulator (Magstim Rapid, MRS1000/50, Magstim Company Ltd., Whitland, UK), TMS (25 Hz, 1000 pulse (biphasic), 10 trains (4 s) of 100 pulses, 60% of the maximum output of the machine) was applied to the 3xTg mice for 4 weeks. All results regarding rTMS are summarized in Table 2.

## 4. Electromagnetic Radiation

Perpendicularly crossed waves of magnetic and electric waves are referred to as electromagnetic fields. These are transverse waves generated by the induction of the electric and magnetic fields. Generally, the speed of an electromagnetic field is approximately 300,000 km/s regardless of the wavelength because it travels without a medium, and higher frequency (or shorter wavelength) electromagnetic waves have higher energy. Based on the wave frequency (Figure 1C), they are categorized into radio waves (1–300 GHz), infrared (300 GHz–400 THz), visible (400–750 THz), ultraviolet (750 THz–30 PHz), and radioactive waves (>30 PHz) [65]. Similar to X-rays, electromagnetic radiation with high energy, which is transmitted through humans, is used to capture internal images in the field of diagnosis. In particular, electromagnetic waves in various forms have been studied for many years as non-invasive treatments for AD, and the potential of electromagnetic radiation for AD treatment has increased. These electromagnetic waves include radio waves, infrared rays, and radioactive rays, which are classified by frequency. The molecular and pathological mechanisms of AD have been identified in various studies using these electromagnetic waves [66,67]. 

### 4.1. Electromagnetic Field Stimulation (EMFS)

EMFS is a non-invasive stimulatory method that uses a radio wave of less than 300 GHz. Over the years, extensive studies have investigated its efficacy on AD symptoms and the molecular/cellular mechanisms underlying its disease-modifying effects, including the clearance of Aβ deposits and tau tangles. These findings have facilitated new techniques for AD treatment. Depending on the wave frequency, electromagnetic fields can be classified into various ranges, including extremely low-frequency electromagnetic fields (ELF-EMFs; <3 kHz), high-frequency electromagnetic fields (HF-EMF; 3–30 MHz), very high-frequency electromagnetic fields (VHF-EMF; 30–300 MHz), and radio-frequency electromagnetic fields (RF-EMF; 20 kHz–300 GHz). In addition, electromagnetic pulse (EMP) stimulation is another stimulatory method of EMFS that has very rapid voltage pulses and up-rising time with a spectral bandwidth range (<1.5 GHz) [68,69]. Generally, electrical stimuli such as tDCS, tACS, and VNS can stimulate a target-specific area depending on the position of the attached electrodes on the skull of the brain. In addition, a magnetic field using TMS can localize a specific brain region depending on the transcranial position of the TMS generator. EMFS enables overall site stimulation because it creates a field containing electromagnetic waves from an antenna source [70]. Because EMFS has a very wide frequency range, from ELF to RF, it can exert various effects depending on the frequency.

As a non-invasive treatment method for AD, EMFS has been continuously investigated from the treatment method to the present. In particular, it has been studied to identify its disease-modifying effects on AD and action mechanisms at the molecular and cellular levels according to variable frequencies in EMF [66,67]. Because the broad frequency spectrum, stimulus intensity, and period are decisive factors in the effect of EMFS, the effect varies depending on the experimental conditions. Therefore, in many studies, investigating the effects of EMFS on the removal of Aβ deposits and tau tangles was the main objective.

In 2007, Giudice et al. investigated the effect of ELF-EMFs on Aβ secretion for the first time [71]. They exposed H4 neuroglioma cells (H4/APPswe) that stably express the human mutant APP to 50-Hz ELF-EMF with an intensity of 3 mT for 18 h. As a result, ELF-EMF stimulation increased total Aβ secretion by 80% in H4/APPswe cells compared with an unstimulated group. The assay also exhibited an increase in Aβ42 secretion of approximately 32%. Moreover, another in vivo study investigated the effect of long-term exposure to ELF-EMFs on the pathological symptoms of AD in rats [72]. The SD rats were exposed to 50-Hz ELF-EMF with an intensity of 100 μT, and their stimulation lasted for 12 consecutive weeks. Unexpectedly, no significant changes were observed in the memory ability or Aβ plaque levels in the hippocampus, cortex, or plasma of the ELF-EMF-treated rats compared with a sham group [72].

Recently, Perez et al. investigated the effect of VHF-EMF on Aβ secretion in primary human brain (PHB) cells [73]. For VHF-EMF stimulation, they used a frequency of 64 MHz, calculated as the minimum power to obtain the minimum specific absorption rate (SAR) that has a biological effect on cells. The cultured PHB cells were stimulated using 64-MHz EMF for 1 h for 14 days based on a SAR of 0.6 W/kg. They observed that EMF treatment reduced the levels of Aβ40 (46%) and Aβ42 (36%) compared with those of the control cells. When based on a SAR of 0.4 W/kg with the same conditions, the levels of Aβ40 and Aβ42 of EMF were significantly reduced by the stimulation of EMF without cell toxicity, whereas no change occurred in the level of sAPPα. Moreover, they further compared the effect of Aβ levels after the treatment at 64 MHz with a SAR of 0.9 W/kg for 4 or 8 days to the cultured PHB cells. Four days of EMF treatment resulted in a significant decrease in Aβ40 levels, whereas no significant difference occurred in the level of Aβ42. Meanwhile, the levels of Aβ40 and Aβ42 were decreased through EMF treatment for 8 days.

In addition, studies have investigated the influence of RF-EMF at cell phone frequencies (800–1000 MHz). For the cellular level, Tsoy et al. verified the stimulatory effect with 217-Hz pulsed 918-MHz EMF, and the medium SAR was set at 0.2 W/kg for 60 min of stimulation [74]. In cultured human and rat astrocytes, RF-EMF stimulation significantly reduced the levels of cellular and mitochondrial oxidative stress generated by the application of Aβ42 and hydrogen peroxide. Moreover, Arendash et al. conducted an in vivo study to verify the efficacy of 918-MHz EMF in AβPPsw transgenic mice (21–27 months old) [75]. The mice were stimulated using 217-Hz pulsed 918-MHz EMF with a SAR of 0.25–1.05 W/kg for 2-h periods per day for 2 months. After 2 months, they observed that EMF treatment decreased the Aβ burden in the hippocampus (30%) and entorhinal cortex (24%) compared with a sham group and significantly increased the level of soluble Aβ1-40 in the hippocampus and cortex. In addition, they observed a decrease in cerebral blood flow and activation of mitochondrial function after treatment with 918-MHz EMF. 

Furthermore, in vivo studies were conducted to verify the effect of RF-EMF on 5xFAD transgenic mice. For stimulation, EMF was performed at a SAR of 5 W/kg and 1950 MHz five times a week for 2 h a day for a total of 8 months. They observed a decrease in the expression of APP and BACE1 and in the amount of Aβ40 and Aβ42 in the cortex and hippocampus of 5xFAD mice compared with sham control mice. Finally, exposure to RF-EMF for 8 months suppressed Aβ production and reduced Aβ deposition. In contrast, another study reported a non-significant disease-modifying effect of Aβ in 5xFAD transgenic mice [76]. Despite the same conditions of RF-EMF above, they were exposed for 3 months and consequently had a negligible effect on the accumulation of Aβ, APP, and carboxyl-terminal fragment β [77]. Based on these two studies, the treatment duration for RF-EMF may be a critical consideration.

Recently, the efficacy of RF-EMF was investigated in five patients with AD [78]. Patients with AD wore head caps with EMF devices. For stimulation, they were exposed once or twice a day for 1 h at 217 Hz and 915-MHz RF-EMF of SAR 1.6 W/kg, and sham and EMF treatment was performed in three periods (0–2 months, 10–14 months, and 19–31 months). Consequently, four out of five patients exhibited a significant decrease in Aβ40 and Aβ42 levels in the CSF two months after EMF treatment. Moreover, a decreased level of pTau was observed in the CSF after 14 months of treatment, with decreases of 35%–79% evident in four of the five subjects. In addition, the amount of total tau increased in the CSF, indicating the prevention of pTau oligomer formation [79]. Additionally, RF-EMF is expected to rebalance tau levels by decreasing the pTau and total tau levels in plasma. Ultimately, RF-EMF studies demonstrated the possibility that exposure to an appropriate intensity and duration could remove the brain Aβ burden and reduce the damage that Aβ could cause without side effects.

Jiang et al. investigated the effect of EMP stimulation on Aβ levels in healthy SD rats [68,69]. They irradiated an EMP of 50 kV/m and 100 Hz (pulse repetition frequency) onto the rats, which were divided into sham and experimental groups with the treatment of 100, 1000, and 10,000 pulses (n = 10). In the first trial (2013), they administered a single treatment of 10, 100, 1000, and 10,000 pulses to the rats, and a further investigation (2016) used exposure to 10, 100, 1000, and 10,000 pulses continuously for 8 months. Consequently, they confirmed that EMP exposure increased the protein levels of Aβ and APP. Further mechanistic studies revealed the increased expression of BACE1 and LC3-II in the hippocampus of EMP-treated SD rats. Interestingly, continuous exposure to EMP was observed to induce AD-like behavior in SD rats.

In summary, EMFS has demonstrated new possibilities for the non-invasive therapy of AD using neurostimulation (Table 3). As the therapeutic effect depends on the intensity, frequency, and period of the stimulus, these conditions should be carefully determined. Additionally, further studies are required to elucidate the hidden mechanism at the molecular level and the safety and efficacy of EMFS.

### 4.2. Infrared Light Stimulation 

Infrared light includes the spectrum from the red edge of visible light to the far infrared. This wavelength range is approximately 760–100,000 nm. Infrared rays include near-infrared rays (wavelength: 0.78 to 3 μm) and far infrared rays (50 to 1000 μm). Infrared stimulation is considered to have physiological effects, primarily through photoreceptors [80]. After infrared stimulation, the absorption of infrared light converts signals that can stimulate biological processes that exert therapeutic effects by regulating biological activity [81]. Accumulating evidence indicates the possibility of the treatment of various diseases, including AD. In studies on infrared treatment for AD, the irradiance of infrared rays, time and frequency, and pulse are important factors affecting the therapeutic effect on AD.

In 2013, Grillo et al. demonstrated the possibility of infrared therapy to alleviate the pathological symptoms of AD through a 600 Hz-pulsed infrared stimulation [82]. They treated an old female AD model mouse (TASTPM) for 2 months by placing it in a 5-mW/cm^2^-irradiance 1072 nm infrared LED array on six sides around it. Chronic infrared treatments were conducted for 6-min sessions for two consecutive days, biweekly, for 5 months. As a result, they observed that infrared stimulation significantly reduced Aβ42 plaques and tau tangles in the cerebral cortex of TASTPM mice. Chronic infrared-stimulated mice exhibited reduced protein levels of αB-crystallin, APP, pTau, Aβ40, and Aβ42 (43%–81%). In addition, the stimulation significantly increased the levels of neuroprotective proteins, including heat shock protein (HSP) 60, HSP70, HSP105, and phosphorylated-HSP27 with reduced Aβ cytotoxicity [83].

In addition, the effect of near-infrared light was examined in APP/PS1 and K369I tau (K3) transgenic mice [84]. Transcranial stimulation was performed using 670 nm and 2 mW/cm^2^ light that could reach the cerebellar tissue of the mice [84]. Stimulation was conducted five times a week for 90 s a day and was continued for 4 weeks. Consequently, it was decreased in terms of the levels of burden, size, number per area, and total counts of Aβ plaques in the cerebellar cortex of APP/PS1 mice [84]. Additionally, the stimulation induced a decrease in neurofibrillary tau tangle formation and oxidative stress-induced damage [84].

Recently, Stepanov et al. investigated the therapeutic effect of near-infrared rays at the cellular level for AD [85]. They exposed Aβ treated microglia cells to near-infrared light with an intensity of 30 mW/cm^2^ for 5 min. Near-infrared light treatment alleviated Aβ-induced inflammation by reducing the secretion of inflammatory cytokines and extracellular ROS production [85]. In addition, near-infrared stimulation promotes the restoration of mitochondrial function in glycolytic microglia by Aβ deposition, preventing neuronal death caused by ROS produced by Aβ-altered microglia [85]. 

Far-infrared light can penetrate up to 1.5 inches under the skin, and it has been reported to exhibit various positive biological effects, including ameliorating endothelial dysfunction [86]. Regarding AD, Li et al. conducted a comparative study on the therapeutic effect of visible, near-infrared, and far-ultraviolet rays on AD [87]. They applied visible (500 nm), near-infrared (800 nm), and far-infrared (3000 to 25,000 nm) rays to APP/PS1 mice (8.5 months old), and light irradiation was conducted for 60 min at 0.13 mW/cm^2^ per day for 1.5 months. As a result, stimulation using far-infrared rays resulted in a clear cognitive improvement compared with other treatments. Further investigations determined that far-infrared radiation reduced the Aβ burden in the cortex and hippocampus, whereas it had no effect on the production of Aβ. In addition, far-infrared light alleviated neuronal inflammation in AD mice by activating microglial phagocytosis, restoring the expression of the presynaptic protein synaptophysin, and enhancing the mitochondrial oxidative phosphorylation pathway to increase ATP production, which can induce Aβ clearance. All results regarding infrared light are summarized in Table 4.

### 4.3. Radiation Therapy 

Radiation comprises a group of subatomic particles and electromagnetic waves (photons). Generally, radiation can be divided into two types: ionizing and non-ionizing. EMF and infrared light are non-ionizing radiations with a lower energy range. Ionizing radiation (IR) includes alpha rays, beta rays, gamma rays, and X-rays, which can create electrically charged particles with sufficiently high energy. Interest in the clinical effects of IR exposure on the central nervous system has been increasing. In particular, low-dose IR (LDIR) is prevalent in our living environments, such as cosmic rays, soil radioactivity, and diverse artificial media (CT and X-ray scanning). In the medical field, LDIR has been widely used to diagnose and treat various diseases, including AD. According to a report by National Academies, LDIR ranges from 0 to 0.1 Gy [88]. Radiation therapy is effective and safe under optimal exposure conditions. Several studies have shown that inadequate IR exposure alters the function of the central nervous system by increasing oxidative stress, mitochondrial dysfunction, and proteolysis, thereby reducing neurogenesis, cerebrovascular dysfunction, and aging, resulting in neurodegenerative disorders [89]. Thus, high-doses of IR (HDIR) are known to induce damage to living tissues with morphological alterations and gradual detrimental pathophysiological effects in the brain [90,91,92,93,94,95,96,97]. In addition, several studies have reported harmful effects of LDIR on learning and memory processes [91,92,94,95,98,99]. According to a study by Azizova et al., cumulative doses higher than 0.2 Gy increase the probability of cerebrovascular diseases with Aβ plaque formation in the brain and vessel walls [97]. Moreover, Lowe et al. performed a transcriptome analysis of brain tissue 4 h after whole-body exposure of 0.1 Gy IR (gamma source) to B6C3F1/HSD mice and observed reduced levels of various genes regulating cognitive functions such as glutamate receptor signaling, long-term depression, and potentiation, and vascular damage [99].

Notably, Cherry et al. demonstrated the acceleration of Aβ plaque pathology 6 months after 0.1 and 1 Gy ^56^Fe radiation in APP/PS1 mice and observed an increased level of soluble Aβ (10%–15%) in the cortex and hippocampus [95]. Further investigations indicated that the increased Aβ is attributed to endothelial activation, suggesting altered Aβ trafficking through the BBB. In other words, these results indicate that the increase in Aβ was not due to higher levels of APP and microglial activation. Moreover, 0.5 and 2 Gy ionizing radiation (gamma rays) on cortical neurons increased tau phosphorylation via the oxidative stress-induced activation of c-Jun N-terminal kinase (JNK) and extracellular signal-regulated kinase (ERK) [94]. Rudobeck et al. indicated that 0.1–1.0 Gy proton radiation to the entire body increased Aβ deposition in the cortex of APP/PSEN1 transgenic mice 9 months after irradiation (*p* = 0.034) [93]. Similar to the study by Cherry et al. [95], an increase in Aβ deposition was implied by vascular decrements and impaired transport of Aβ from the brain. Moreover, McRobb et al. reported that 20 Gy X-ray radiation increases the cleavage of APP to Aβ peptides through the downregulation of disintegrin and ADAM10 [92]. Further mechanistic analysis demonstrated that ADAM10 precludes the amyloidogenic pathway by competing with beta-secretase BACE-1. In a Drosophila AD model, irradiation using 4-Gy HIDR (gamma-ray) induced cellular apoptosis through the hyperactivation of p38 mitogen-activated protein kinase (MAPK) signaling pathways, despite the activation of AKT. However, in this study, no difference was observed in the mRNA and protein levels of Aβ after irradiation.

Although some studies have indicated harmful effects, several studies have shown that, as a non-invasive strategy for the treatment of AD, IR exposure can ameliorate the Aβ burden and pTau tangles. A recent experiment demonstrated that 2 Gy radiation to the entire body for five consecutive days reduced various forms of Aβ (soluble peptides, oligomers, fibrils) in the hippocampus of 3xTg-AD mice 8 weeks after irradiation [100]. However, in this study, highly aggregated forms of amyloid (dense core methoxy-X04^32^-positive plaques) were observed [100]. Interestingly, several studies have introduced a mechanism by which radiation therapy acts as a hormone to deliver therapeutic effects through continuous signal transduction. For example, exposure to 0.05 Gy LDIR (gamma ray) suppresses Aβ42-induced cell death by activating the AKT survival signaling pathway and inhibiting the p38 MAPK apoptotic pathway [91]. Khandelwal et al. reported significant decreases in APP, pTau, BACE, and neurofibrillary tangle formations one month after a single 4 Gy gamma irradiation [90]. Further molecular analysis revealed that IR suppresses tau phosphorylation by inhibiting ERK and JNK levels and increasing the inhibitory phosphorylation of GSK-3β at serine 9 (Figure 4A).

Subsequently, Kim et al. described a mechanism for IR-induced inhibition of Aβ accumulation that regulates the microglial phenotype from anti-inflammatory M2 to pro-inflammatory M1 (Figure 4B) [101]. They observed that a radiation dose of 2 Gy per fraction of HDIR (X-ray) to myeloid cells 2 (TREM2) for five times promoted the phagocytic activity of microglia, finally suppressing the pro-inflammatory response. Following this pathway, the authors proposed that Aβ plaques are reduced with an improvement in cognitive impairment 8 weeks after irradiation. Another study reported an increased number of microglia 4 weeks after long-term radiation treatment (10 Gy for five fractions), which further increased the number of microglia after 8 weeks [102]. Moreover, another study reported that 3 Gy in five fractions of LDIR reduces the activation of microglia, whereas its stimulation enhances phagocytic activity by reactive microglia near amyloid plaques, finally resulting in a decrease in the size of amyloid plaques without a change in the total number of Aβ [103]. Wilson et al. observed a significant decrease in Aβ burden and tau tangles in 3xTg-AD mice 8 weeks after the last exposure to five fractions of 2 Gy LDIR [104]. These results indicated that the neuroprotective effects are attributed to the inhibition and decrease in amyloid plaques and improvement of cognitive function by regulating the phenotype of microglia involved in the phagocytosis of abnormal proteins in brain tissue, synapse formation, and synapse pruning. Interestingly, disease-modifying effects on AD were observed at least 8 weeks after irradiation.

However, several studies have also reported that radiation therapy has no effect on Aβ accumulation [105,106,107]. A total of 9 Gy (1.8 Gy per fraction for five fractions) of HDIR failed to affect 5xFAD mice 4 days after irradiation [107]. In addition, 10 Gy (2 Gy fractions for 5 days daily) and the same dose for 5 weeks of weekly IR indicated no impact on the density of amyloid plaques 4 months after irradiation [106]. In contrast to a previous study [95], there was no change in the plaque and tau pathology after irradiation with iron or silicon in the range of 0.1 to 2 Gy one day after irradiation [105]. 

To summarize the results of the above-mentioned experiments, in most cases, radiation exposure does not mediate a direct effect on Aβ; however, it indirectly affects it by regulating the number and/or activation of microglia and anti-inflammatory cytokines. Ultimately, the optimal conditions for radiation required to treat patients with AD may depend on the results of the ongoing clinical studies. Therefore, additional clinical and preclinical studies are required to determine the optimal total dose for AD treatment.

## 5. Transcranial Focused Ultrasound

Thus far, the effects of transverse waves, such as tDCS, TMS, and EMFS, on β-amyloid plaques and tau tangles (pathological molecular markers of AD) have been described. This section introduces a longitudinal wave that oscillates parallel with the direction of wave progress. Ultrasound is a representative longitudinal waveform with higher frequencies than the upper range of human audible limits (20 kHz) (Figure 1D). Because ultrasound has unique attributes, such as reflection, scattering, and absorption, it has been used for diagnostic imaging, thermal ablation, and drug delivery. Recently, the pathological hallmarks of AD, such as Aβ plaques and tau tangles, have been targeted using sonication treatment. Transcranial-focused ultrasound is a non-invasive therapeutic technique that uses ultrasonic waves of low intensity (<0.72 W/cm^2^) and low frequency (<2 MHz), compared with ultrasound for imaging (2–20 MHz). Transcranial-focused ultrasound modalities include MRI-guided focused ultrasound (MRgFUS), scanning ultrasound, and low-intensity focused ultrasound (LIFU). These ultrasonic stimulations facilitate the opening of the BBB for drug delivery (particularly for Aβ antibodies and nanoparticles), and this BBB opening effect is known to be enhanced when microbubbles are injected intravenously (Figure 5). In addition, the stimulation of focused ultrasound is known to regulate the production and clearance of Aβ plaques and tau tangles through the activation of glial cells such as microglia and astrocytes in the brain.

### 5.1. Focused Ultrasound

Although some studies have performed co-treatment with the injection of microbubbles and/or a certain therapeutic agent to improve the effects of ultrasound, ultrasound treatment can result in a single-handed reduction of Aβ plaques and improvement of impaired memory in AD-related mice. For instance, three repetitive exposures of LIFU (1 MHz, 0.528 W/cm^2^, 50 ms bursts, 1 Hz pulse repetition frequency (PRF)) were conducted for 5 min in each session for 42 days to the hippocampus of an aluminum chloride-induced rat model of AD [108]. Consequently, LIFU stimulation resulted in a decrease in Aβ42 expression, improving memory retention and memory deficits [108]. Subsequently, Eguchi et al. demonstrated that LIFU stimulation (1.875 MHz, 0.099 W/cm^2^, 0.017 ms bursts, three times of 20 min for a total of 11 of 86 days) to the entire brain ameliorates impaired cognitive function in 5xFAD transgenic mice with a decrease in Aβ plaques [109]. Further RNA sequencing analysis revealed that endothelial nitric oxide synthase (eNOS) associated with glial cell activation is upregulated by LIFU stimulation [109]. Interestingly, Bobola et al. compared the effects of acute (for 1 h) and chronic (1 h per day for 5 days) stimulation of focused ultrasound (2 MHz, 40 Hz pulse repetition frequency, 400-μs-long pulses, 3.0 W/cm^2^) on the brain hemisphere of 5xFAD transgenic AD mice. As a result, the acute application of focused ultrasound caused an increase in activated microglia that colocalized with Aβ plaques compared with other control hemispheres, and chronic application further resulted in a reduced Aβ burden compared with a sham control group [110]. However, Leinenga et al. demonstrated that scanning ultrasound stimulation (1 MHz, 0.7 MPa, 10 ms bursts, 6 s per spot, 10-Hz PRF, weekly for 5 weeks) to one brain hemisphere fails to significantly reduce the amyloid burden, including plaque size and number [111]. 

#### 5.1.1. Focused Ultrasound with Microbubbles Infusion (FUS-MB)

The delivery of large therapeutic drugs (antibodies, proteins, gene therapeutics, and stem cells) has been difficult owing to physical barriers to the brain, such as BBB, blood–CSF barriers, and arachnoid barrier [112]. To overcome this obstacle, scholars have studied focused ultrasound coupled with the infusion of microbubbles (FUS-MB) in recent years to verify its effects on Aβ plaques and tau tangles through BBB opening [113,114,115,116,117,118,119]. Generally, focused ultrasound can induce the oscillation of microbubbles about 1–10 µm in diameter in the focal vasculature, which is known to increase BBB permeability by more than 100-fold. To facilitate this effect, numerous studies have attempted to exogenously supply microbubbles into veins.

MRgFUS is a focused ultrasound that uses MRI to measure the extent of the BBB opening size and monitor the side effects after ultrasound exposure. Through this method, Burgess et al. observed an increased BBB permeability in the bilateral hippocampus (1.68 MHz, 10-ms bursts, 120 s duration, 1-Hz PRF, weekly for 3 weeks) after the treatment of focused ultrasound with an intravenous dose of 0.02 mL/kg of body weight of microbubbles, and they demonstrated a significant decrease in the number of Aβ plaques and their size in 7-month-old transgenic (TgCRND8) mice [118]. Moreover, repeated scanning ultrasound treatments with microbubble injection caused the Aβ plaque to internalize into the activated microglia in APP23 transgenic mouse brains; interestingly, no change in the number of microglia was observed [120]. Ultrasound-treated mice restored the impaired memory observed in memory tasks such as the Y-maze, novel object recognition test, and active place avoidance task [120]. Activated microglia engulfing Aβ deposits in response to LIFUS-MB stimulation were consistently observed in a 3xTg-AD mouse model [121]. Furthermore, Poon et al. demonstrated that the effect of focused ultrasound-treated Aβ plaque reduction persisted for two weeks [116]. Based on this result, they observed that three to five biweekly treatments in TgCRND8 mice caused a significant improvement in plaque number and size compared with untreated littermates [116]. Recently, Lee et al. reported that treatment with FUS-MB removes the Aβ burden in 5xFAD mice by improving glymphatic-lymphatic drainage, which is a waste clearance pathway in the brain [113]. 

The application of FUS-MB has also been shown to reduce pathological tau in AD mouse models, including rTg4510 and K369I tau transgenic mice. In this regard, this effect of tau tangles was observed to be driven by the migration of resident microglia and the infiltration of peripheral immune cells through the BBB opening toward tau tangles [115]. In addition, the autophagy pathway in hippocampal neurons increases after SUS-MB treatment [114,122].

In a clinical study, Lipsman et al. demonstrated the transient opening of the BBB in response to focused ultrasound coupled with injected microbubbles in five patients with AD using the MRIgFUS system [123]. Thereafter, another study further supported the safety, feasibility, and reversibility of BBB opening with focused ultrasound treatment of the hippocampus and entorhinal cortex in six patients with AD [124]. Further investigation showed a remarkable decrease in Aβ deposits of 5.05% (±2.76) in six patients with AD [125]. However, there was no effect of implantable stimulation with focused ultrasound in patients with AD. Epelbaum et al. attempted to implant a 1-MHz ultrasound device in the skull of 10 patients with mild AD [126]. For 9 months, the volunteered patients were stimulated in seven ultrasound sessions (1-MHz, 0.9–1.03 MPa, 4 min duration, 1-Hz PRF) with intravenous infusion of microbubbles twice per month [126]. However, they failed to prove a significant difference in the FUS-MB stimulation of β amyloid accumulation and cognitive recognition [126]. 

#### 5.1.2. Focused Ultrasound with Microbubble Infusion and Drug Delivery

Numerous studies have demonstrated ultrasound-mediated BBB opening in animal AD models and clinical patients with AD. Undoubtedly, this BBB opening is expected to burst Aβ clearance in the brain when co-treated with additional therapeutic agents such as anti-Aβ antibodies. Thus, many investigations have stimulated focused ultrasound with microbubble injections and therapeutic agents.

In this regard, Raymond et al. attempted to determine the delivery of intravenously injected Aβ antibody after stimulation using focused ultrasound (0.69 MHz, 0.67–0.8 MPa, 10 ms bursts, 40–45 s duration, 1-Hz PRF) with microbubble injection under MRI guidance. As a result, they observed increased Aβ antibodies (2.7 ± 1.2-fold) in Aβ plaque-monitored brain regions [127]. Further studies have shown that single or repeated treatment with FUS-MB enhanced the permeability of intravenously administrated Aβ antibodies into the target region of the brain, which ultimately ameliorated the Aβ plaque and cognitive impairments in several AD-related mice [128,129,130,131,132]. In addition, treatment with FUS-MB was observed to increase the delivery of endogenous immunoglobulins within the focused ultrasound-treated cortex (0.5 MHz, 0.3 MPa, 10 ms bursts, 120 s duration, 1-Hz PRF), which caused the activation of microglia and astrocytes involved in the internalization of Aβ [119]. 

When comparing the stimulation of single and multiple sessions (two and three sessions), the multiple ultrasound treatments of MRIgFUS (1 MHz, 0.8 MPa, 10 ms bursts, 20 s duration, 1-Hz PRF) with anti-Aβ antibody (BC-10) exhibited a more effective clearance of Aβ burden than that of a single session [131]. In addition, repeated stimulations of focused ultrasound such as SUS-MB (1-MHz, 0.7 MPa, 10 ms bursts, 6 s per spot, 10-Hz PRF) [133] and FUS-MB (0.4 MHz, 0.41–0.5 MPa, 10 ms bursts, 60 s duration, 1-Hz PRF) [134] were observed to deliver other larger molecules such as tau antibodies (29–156 kDa), glycogen synthase kinase-3 (GSK-3) inhibitor (308 kDa), and nanoparticles (Qc@SNPs) [135]. In these experiments, repeated SUS-MB treatment with tau antibodies caused a significant decrease in pTau by promoting the delivery of tau antibodies (RN2N) to the brain [133,136]. Repeated FUS-MB stimulation enhances the permeability of the GSK-3 inhibitor, which is involved in the amelioration of Aβ plaques [134]. Xu et al. [137] and Liu et al. [135] further reported that FUS-MB stimulation enhanced the delivery of nanocarriers such as protoporphyrin IV-modified oxidized mesoporous carbon nanospheres (PX@OP@RVGs) and quercetin-modified sulfur nanoparticles (Qc@SNPs) to a specific target region, which exhibited a significant reduction of Aβ plaques and pTau tangles. Recently, the stimulation of FUS-MB (1 MHz, 10 ms bursts, 200 s duration, 1-Hz PRF) with Gastrodin (phenolic glycoside extracted from the Chinese herb) injection alleviated AD-related memory deficits, resulting in reduced levels of Aβ burden and pTau in the hippocampus [138]. In addition, several studies have used passive cavitation detection (PCD) to control the acoustic pressure within a safe range, which maintains the acoustic pressure when sub-harmonic emission is detected [130,135,138,139]. 

Stimulation using focused ultrasound induces BBB opening, enabling various therapeutic molecules to permeate the brain. Because the skull of humans is much thicker than that of animals such as rodents and rabbits, a higher-power focused ultrasound is used to induce BBB opening in human studies. The higher power is occasionally likely to cause tissue damage as an undesired effect, e.g., an animal study reported a hemorrhage after the ultrasound stimulation with acoustic pressure of 0.67 and 0.8 MPa [127,131]. To determine the safety and efficacy, acoustic pressure should be sensitively controlled with other parameters, including transducer frequency, burst lengths, PRF and administrated method, duration of microbubbles, and therapeutic agents. Future studies can use a passive cavitation detector to monitor cavitation in real-time and adjust the acoustic pressure threshold to exhibit therapeutic effects. All results regarding focused ultrasound are summarized in Table 5.

## 6. Conclusions and Perspectives

In this review, we discuss the disease-modifying effects (particularly focused on Aβ plaques and tau tangles) of non-invasive electroceuticals that are categorized by energy forms such as electric waves, magnetic waves, electromagnetic waves, and ultrasound. 

Currently, electroceuticals are being used clinically in various fields. For example, electrical stimulation such as tDCS, tACS, and VNS is used to relieve pain, migraine, and developmental disability. Moreover, magnetic field stimulation such as TMS is used for areas such as depression. Infrared stimulation is used to treat hair loss. In addition, electromagnetic waves, ionizing radiation, and ultrasound are being studied for use in treatment beyond diagnosis. For this purpose, studies are being conducted with a focus on pathological molecules.

To date, numerous studies have demonstrated the disease-modifying effect of Aβ plaques and tau tangles through brain stimulation using electroceutical methods. However, some studies have reported negative or no effects after stimulation with electroceuticals. These controversial effects are expected to be attributed to different stimulatory and/or experimental conditions. Thus, further investigations are required for optimizing and standardizing the variable conditions of electrostimulation, such as the electrode position, intensity, frequency, duration, and treatment session, to maximize the therapeutic effect for practical applications. In addition, stimulatory devices should be improved to achieve functionality based on interdisciplinary biomedical engineering and medical science.

## Figures and Tables

**Figure 1 ijms-24-00679-f001:**
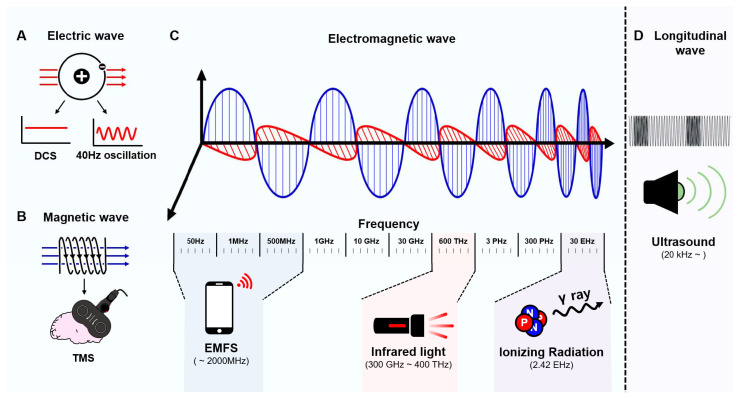
Brief overview of power source of electroceuticals used for the treatment for AD. (**A**) An electric wave is divided into direct current and alternative current, which are used for DCS and 40-Hz oscillation, respectively. (**B**) Magnetic field-induced wave used for the stimulation of TMS. (**C**) Perpendicularly crossed waves of magnetic and electric waves are referred to as electromagnetic fields, which are used for EMFS (~2000 MHz), stimulation of infrared light (300 GHz–400 THz), and stimulation of ionizing radiation (2.42 EHz). (**D**) Ultrasound is a longitudinal waveform used for the stimulation of focused ultrasound, which requires a medium for transferring the waveform.

**Figure 2 ijms-24-00679-f002:**
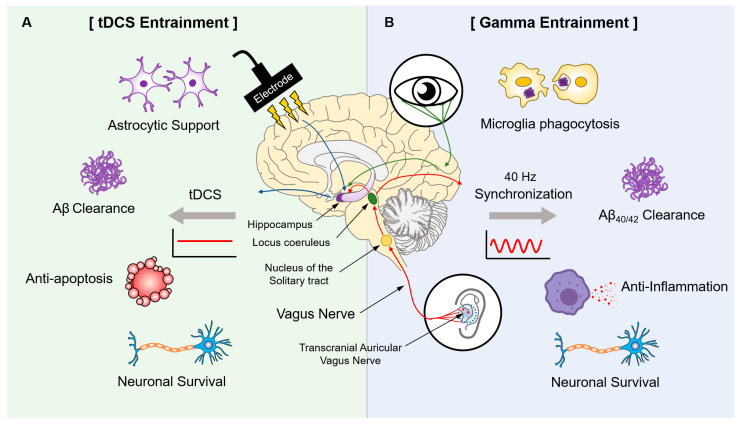
Therapeutic effects of electrical waves such as tDCS and gamma entrainment in AD. (**A**) Stimulation using tDCS results in the astrocytic support, Aβ clearance, anti-apoptosis, and a neuronal survival effect in the brain after exposure. (**B**) The transcranial auricular vagus nerve and audiovisual sensory nerve are stimulated for 40-Hz gamma oscillation, which activates microglia phagocytosis, Aβ clearance, anti-inflammation, and increased neuronal survival.

**Figure 3 ijms-24-00679-f003:**
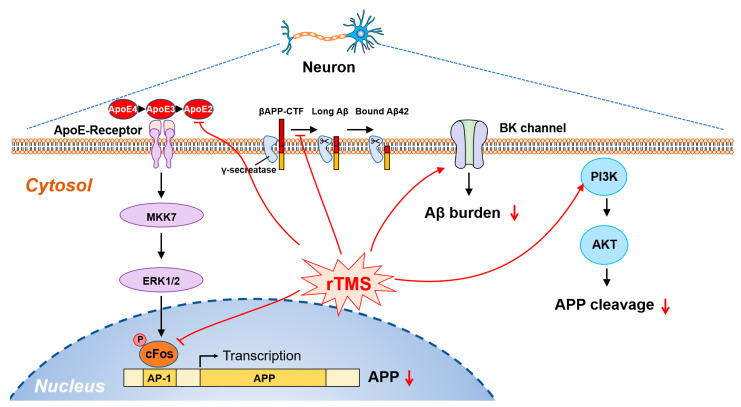
Schematic of signaling pathways responding to the exposure of rTMS in AD. The treatment of rTMS suppresses the transcriptional expression of APP through the inhibition of the MKK7-ERK1/2-cFOS axis and facilitates Aβ clearance by decreasing the expression of ApoE. Moreover, rTMS activates the BK channel and PI3K, resulting in a decrease in the Aβ burden and APP cleavage, respectively.

**Figure 4 ijms-24-00679-f004:**
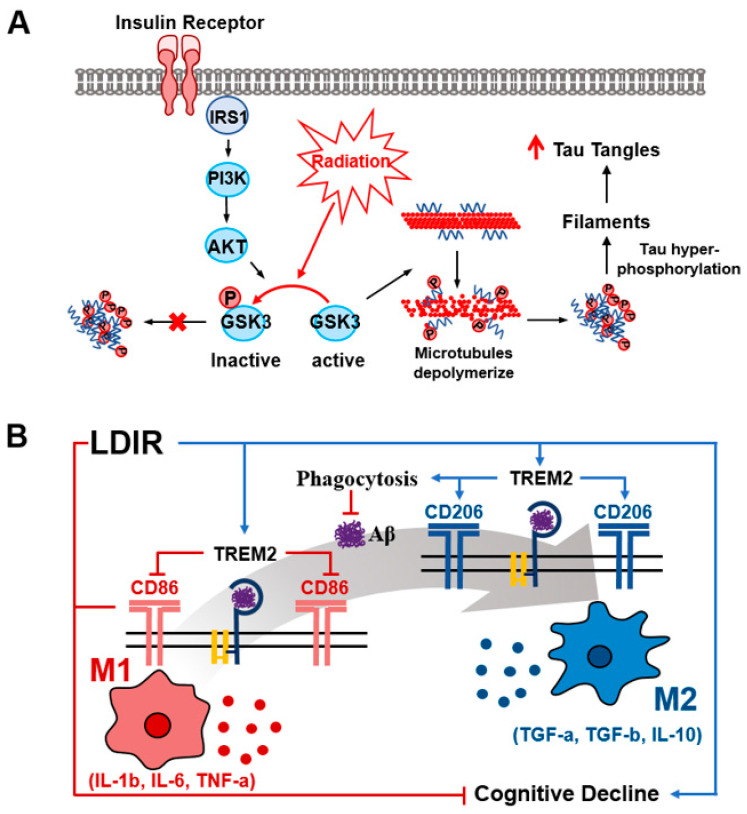
Schematic of signaling pathways and dose/time-dependent effects of ionizing radiation. (**A**) Schematic representation of irradiated dose-dependent biphasic effects at the molecular level. A high dose irradiation inhibits the activity of GSK-3β, thereby decreasing the tau tangles. (**B**) Schematic representation of the modulatory effects of LDIR exposure on the shifting phenotype of the macrophage.

**Figure 5 ijms-24-00679-f005:**
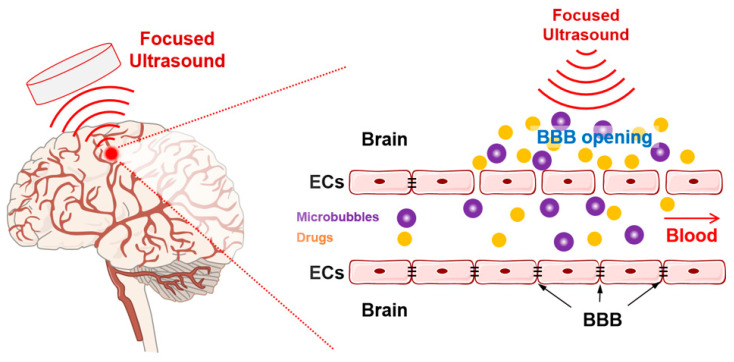
Conceptual schematic representation of therapeutic effects of focused ultrasound. Focused ultrasound induces the opening of the blood–brain barrier through microbubble generation, and it can be coupled with microbubble injection and drug delivery for improved efficacy.

**Table 2 ijms-24-00679-t002:** A summary of rTMS on Aβ and tau pathology.

Type of rTMS	Stimulation Intensity and Frequency	Duration	Subject	Main Finding	Reference
Chronic TMS	80% of the maximum output of machine (MRS1000/50) 1, 10, 15 Hz	Pulse uprise time: 60 μs Pulse duration: 250 μs 5 s once a day for 4 weeks	3xTg mice	- ↑ Homer1a expression - ↑ BK channel activity - ↓ Aβ42 level - ↑ hippocampal LTP	[52]
	1.26 T, 2 s inter-train interval, 1 and 10 Hz	14 consecutive days	Aβ1-42 injected mice	- ↓ GSK-3β and phosphorylation of tau - ↑ β-catenin signaling - ↑ survival of neurons	[56]
	1.26 T, 2 s inter-train interval, 1 and 10 Hz	14 consecutive days	6-OHDA-induced mice	- ↓ MKK7-ERK1/2-c-FOS-APP axis - ↓ Amyloid precursor protein level - ↓ Aβ42 level in brain tissues - ↑ cognitive behaviors	[59]
	120% of the average resting motor threshold (CCY-3) Total 600 pulses (20 bursts, 30 pulses each train) 2 s inter-train interval, 5 Hz	14 consecutive days	APP/PS1 mice	- ↓ ApoE, p62 and PP2A expressions - ↑ Aβ, APP and pTau levels - ↑ hippocampal autophagy - ↑ spatial, cognitive learning and memory	[62]
	1.38 T, 100 sessions (40 bursts trains), 5 s inter-session interval 20 Hz	14 consecutive days	5xFAD mice	- ↓ active glial cells (microglia and astrocytes) - ↑ neuronal activity	[63]
	60% of the maximum output of machine (MRS1000/50) Total 1000 pulses (10 trains of 100 pulses) 25 s inter-train interval, 25 Hz	28 consecutive days	3xTg mice	- ↓ hippocampal Aβ42 levels - ↑ PI3K and Akt signals - ↑ alleviation of cognitive deficits	[64]

**Table 3 ijms-24-00679-t003:** A summary of EMF stimulation on Aβ and tau pathology.

Type of EMF	Stimulation Intensity and Frequency	Duration	Target Tissues or Animals	Main Finding	Reference
ELF-EMF	3.1 mT (alternating magnetic field), 50 Hz	18 h 1 session	H4 neuroglioma cells (H4/APPswe)	- ↑ total Aβ level 80% - ↑ Aβ42 level 32%	[71]
	100 μT (alternating magnetic field), 50 Hz	continuously 12 weeks	Sprague Dawley rats	- Insignificant change in Aβ level	[72]
VHF-EMF	0.4 W/kg, 0.6 W/kg, 0.9 W/kg, 64 MHz	1–2 h 4–14 days	Primary human brain (PHB) cells	- ↓ significantly in Aβ40 and Aβ42 levels	[73]
RF-EMF	0.20 W/kg, 918 MHz, 217 Hz pulse	1 h 1 session	primary human and rat astrocytes	- ↓ ROS from Aβ42 and H2O2 - ↓ mitochondrial ROS formation induced by Aβ42 - ↓ activity of NADPH-oxidase by Aβ42 - ↑ mitochondrial membrane potential	[74]
	0.25–1.05 W/kg, 918 MHz, 217 Hz pulse	2 h 12 days or 2 months	AβPPsw (Tg) mice	- ↓ Aβ burdens 30% in hippocampus and 24% in entorhinal cortex - ↑ significantly level of soluble Aβ1-40 in hippocampus and cortex	[75]
	1.6 W/kg, 915 MHz	once or twice 1 h 2, 4, 12 months	5 AD patents	- ↓ CSF levels of C-reactive protein, pTau217, Aβ40, and Aβ42 - modulating CSF oligomeric Aβ levels	[78]
	5 W/kg, 1950 MHz	2 h, 5 days per week 8 months	Tg-5xFAD mice	- ↓ ratio of Aβ42 and Aβ40 - ↓ APP and BACE1 expression - ↓ reduction of neuroinflammation	[76]
		2 h, 5 days per week 3 months		- no affect in accumulation of Aβ - no affect in level of APP and CTFβ	[77]
EMP	50 kV/m, 100 Hz	100, 1000, 10,000 pulses 2 months	Sprague Dawley rats	- ↑ Aβ protein level - ↑ expression of APP and Aβ oligomer	[68]
				- ↑ Aβ protein and oligomer - ↑ BACE1 and aberrant cleavage of APP	[69]

**Table 4 ijms-24-00679-t004:** A summary of infrared light stimulation on Aβ and tau pathology.

Type of Infrared Light	Stimulation Intensity and Frequency	Duration	Target Tissues or Animals	Main Finding	Reference
Pulsed Infrared	5 mW/cm^2^, 1072 nm pulse: 600 Hz (duty cycle 0.3 ms)	6 min, two consecutive days biweekly for 5 months	Female TASTPM mice	- ↓ significantly in Aβ_1-42_ plaques in the cerebral cortex - ↓ αB-crystallin, APP, pTau, Aβ40 and Aβ42 (43–81%) protein expression - ↑ significantly in HSP60, 70 and 105 and phosphorylated-HSP27	[82]
Near infrared	2 mW/cm^2^, 670 nm	90 s, day, 5 days 4 consecutive weeks	APP/PS1 transgenic mouse and K3 mouse	- ↓ Aβ plaque burden in cerebellar cortex in APP/PS1 mice - ↓ formation of neurofibrillary tangles and hyperphosphorylation of tau - ↓ damage caused by oxidative stress	[84]
	30 mW/cm^2^, 808 nm	5 min 1 session	Aβ treated microglia cells	- ↑ phagocytosis activated microglia level - ↓ inflammatory cytokines and extracellular ROS production - ↓ death of neurons caused by Aβ-altered microglia	[85]
Far infrared	0.13 mW/cm^2^, 3000–25,000 nm	1 h 1.5 months	APP/PS1 mice	- ↓ Aβ burden in cortex and hippocampus - ↓ neuroinflammation by activating microglia’s phagocytosis - ↑ ATP production and expression of the presynaptic protein synaptophysin - ↑ learning and memory ability	[87]

**Table 5 ijms-24-00679-t005:** A summary of ultrasound on Aβ and tau pathology.

Type of Ultrasound	Stimulation Intensity and Frequency		Duration		Subject	Main Finding	Reference
FUS	Type: LIPUS Frequency: 1 MHz $I_{SPTA}$$=0.528\;W/{\mathrm{cm}}^{2}$ Burst length: 50 ms PRF: 1 Hz Duty cycle: 5%		Sonication duration: 5 min with triple sonication daily for 42 daays		$A1{\mathrm{Cl}}_{3}$ treated mice	- ↓ protein expression of Aβ content - ↓ memory retention and memory deficits - ↑ neurotrophic factors controlling or reversing AD	[108]
	Type: LIPUS Frequency: 1.875 MHz $I_{SPTA}$$=0.099\;W/{\mathrm{cm}}^{2}$ Burst length: 0.017 ms		Sonication duration: 20 min with triple sonication on days (1, 3, 5, 28, 30, 32, 56, 58, 60, 84)		5XFAD mice	- ↓ Expression of APP and BACE-1, changes in characteristics of microglia and reduce Aβ - ↓ cognitive dysfunction by reducing Aβ and microgliosis	[109]
	Type: SUS Frequency: 1 MHz Acoustic pressure = 0.7 MPa Burst length: 10 ms PRF: 10 Hz Duty cycle: 10%		Sonication duration: 6 s per spot treated weakly for 5 weeks		APP23 mice	- insufficient in amyloid clearance - ↓ reductions in synaptic activity	[111]
	Type: LIPUS Frequency: 2 MHz $I_{SPTA}$$=3.0\;W/{\mathrm{cm}}^{2}$ Burst length: 0.4 ms PRF: 40 Hz Duty cycle: 5%		Sonication duration: 1 h single treatment/repeated treatment daily for 4 days		5XFAD mice	- ↑ activation of microglia - ↓ reduction in Aβ burden	[110]
FUS-MB	Type: MRIgFUS Frequency: 1.68 MHz Acoustic pressure: When sub-harmonic emissions were detected, the acoustic pressure was reduced to half and maintained for the remainder of sonication duration Burst length: 10 ms PRF: 1 Hz MB: Definity 0.02 mL/kg		Sonication duration: 120 s Weekly for 3 weeks		TgCRND8 mice	- ↓ significantly in Aβ - ↑ astrocytes and microglia which internalized amyloid - ↑ production of BDNF - ↑ Akt signaling	[118]
	Type: SUS Frequency: 1 MHz Acoustic pressure: 0.7 MPa Burst length: 10 ms PRF: 10 Hz Duty cycle: 10% MB: in-house Lipid-shelled MB 0.001 mL/g		Sonication duration: 6 s per spot Weekly for 6 or 7 weeks		APP23 mice	- ↑ microglia Aβ lysosomal activity - ↑ albumin entering the brain, which binds to Aβ and facilitates Aβ uptake by microglia - ↑ memory ability	[120]
	Type: Intracranial FUS Frequency: 1.1 MHz In situ pressure: 0.4–0.8 MPa Burst length: 10 ms PRF: 1 Hz MB: Definity 0.04 mL/kg	Type: MRIgFUS Frequency: 1.68 MHz Acoustic pressure: When sub-harmonic emissions reached threshold of 3.5 times the magnitude of background signals, the acoustic pressure was reduced by 50% and maintained for the remainder of sonication duration Burst length: 10 ms PRF: 1 Hz MB: Definity 0.02 mL/kg	Sonication duration: 120 s Single treatment	Sonication duration: 120 s Biweekly for 10 weeks	TgCRND8 mice	- ↑ infiltration of systemic phagocytic immune cells into the brain - ↑ phagocytosis of Aβ in microglia and astrocytes - ↑ entry of endogenous immunoglobulins which binds to Aβ plaque	[116]
	Type: MRIgFUS Frequency: 220 kHz Acoustic pressure: When sub-harmonic emissions were detected, the acoustic pressure was reduced to half and maintained for the remainder of sonication duration Burst length: 2 ms on 28 ms off for 300 ms Repetition interval: 2.7 s MB: Definity 0.004 mL/kg		Sonication duration: 50 s Two treatment sessions with 1 month interval		Five 50–85 years old AD patients	- ↑ BBB open - no significant changes in cognition or functioning	[123]
	Type: FUS Frequency: 1.5 MHz Acoustic pressure: 0.45 MPa Burst length: 6.7 ms PRF: 10 Hz MB: in-house MB 0.0001 mL/g		Sonication duration: 60 s Weekly for 4 weeks		rTg4510 mice	- ↑ activation of microglia f infiltrating immune cells that help reduce pTau - ↑ migration of resident microglia - ↑ infiltration of peripheral immune cells through the BBB opening	[115]
	Type: SUS Frequency: 1 MHz Acoustic pressure: 0.25 MPa Burst length: 10 ms PRF: 1 Hz MB: Definity		Single treatment		K3691 tau transgenic mice	- ↓ active glial cells (microglia and astrocytes) - ↑ autophagy in neurons which contributes to tau clearance	[115]
	Type: FUS Frequency: 0.715 MHz Acoustic pressure: 0.42 MPa Burst length: 20 ms PRF: 1 Hz Duty cycle: 2% MB: SonoVue 0.1 mL		Sonication duration: 60 s Weekly for 6 weeks		5xFAD mice	- ↑ clearance of Aβ via glymphatic-lymphatic system - ↑ restoration of memory via glymphatic-lymphatic clearance of amyloid	[114]
	Type: MRIgFUS Frequency: 220 kHz MB: Definity		Three treatment sessions with 2 weeks interval		Six early AD patients	- ↑ BBB open - no significant changes in clinical aspect	[124]
	Type: implantable ultrasound device (SonoCloud-1) Frequency: 1 MHz Acoustic pressure: 0.9 MPa–1.03 MPa MB: SonoVue 0.1 mL/Kg		Sonication duration: 4 min Every 2 weeks for 3.5 months		10 AD patients	- no significant effect in Aβ accumulation and cognitive recognition	[126]
FUS-MB with drug delivery	Type: MRIgFUS Frequency: 0.558 MHz Acoustic pressure: 0.3 MPa Burst length: 10 ms PRF: 1 Hz MB: Definity 0.16 mL/Kg		Sonication duration: 120 s Single treatment		TgCRND8 mice	- ↑ entering of anti-Aβ antibody into the brain - ↓ Aβ plaque burden (size and total surface area)	[129]
	Type: MRIgFUS Frequency: 1 MHz Acoustic pressure: 0.8 MPa Burst length: 10 ms PRF: 1 Hz MB: SonoVue 0.05 mL/kg		Sonication duration: 20 s Single treatment 3 repeated sessions with 3 days respectively		2% high cholesterol diet New Zealand White rabbits	- ↑ entering of anti-Aβ antibody into the brain - ↑ anti-Aβ antibody	[131]
	Type: SUS Frequency: 1 MHz Acoustic pressure: 0.7 MPa Burst length: 10 ms PRF: 10 Hz Duty cycle: 10% MB: in-house Lipid-shelled MB 0.03 mL		Sonication duration: 6 s per spot Weekly for 4 weeks		pR5 mice	- ↓ interaction between GSK-3β and tau, thereby tau phosphorylation is prevented - ↓ phosphorylated tau levels - ↑ RN2N delivery across the BBB	[133]
	Type: MRIgFUS Frequency: 0.5515 MHz Acoustic pressure: based on the analysis of MB signal recording during each burst Burst length: 10 ms PRF: 1 Hz MB: Definity 0.04 mL/Kg		Sonication duration: 120 s Single treatment		TgCRND8 mice	- ↑ entering of BAM-10 into the brain, inducing clearance of Aβ	[135]
	Type: MRIgFUS Frequency: 0.69 MHz Peak negative pressure: 0.67–0.8 MPa Burst length: 10 ms PRF: 1 Hz		Sonication duration: 40–45 s		B6C3-Tg mice	- ↑ entering endogenous IgG and anti- Aβ antibodies - ↑ Aβ plaque clearing	[127]
	Type: FUS Frequency: 0.4 MHz Acoustic pressure: 0.41–0.5 MPa Burst length: 10 ms PRF: 1 Hz MB: SonoVue 0.01 mL/kg		Sonication duration: 60 s Single treatment 7 days (Exposure for a total 5 times)		APPswe/PSEN1-dE9 mice	-↑ GSK-3 inhibitor (AR-A014418) - ↓ Aβ significantly	[134]
	Type: FUS Frequency: 1 MHz Acoustic pressure: 0.41–0.5 MPa		Sonication duration: 180 s Single treatment		APP/PS1 mice	- ↑ delivery of Nanoparticles (PX@OP@RVG) into the brain - ↓ Aβ plaque and phosphorylated tau	[137]
	Type: SUS Frequency: 1 MHz Acoustic pressure: 0.65 MPa (for whole brain) /0.6 MPa (for hippocampus) Burst length: 10 ms PRF: 10 Hz Duty cycle: 10% MB: in-house Lipid-shelled MB 0.04 mL		Sonication duration: 6 s per spot (for whole brain) 60 s (for hippocampus)		pR5 mice	- ↑ various formats delivery of anti-tau antibody	[136]
	Type: MRIgFUS Frequency: 1.68 MHz Acoustic pressure: when a 840 Hz sub-harmonic was detected, the pressure amplitude was dropped to 50% and maintained for the remainder of sonication duration Burst length: 10 ms PRF: 1 Hz MB: Definity 0.02 mL/Kg		Sonication duration: 120 s Single treatment		TgCRND8 mice	- ↑ delivery of recombinant adeno-associated virus mosaic serotype - Regulate transgene expression near Aβ plaque	[139]
	Type: FUS Frequency: 0.4 MHz Acoustic pressure: 0.41–0.5 MPa Burst length: 10 ms PRF: 1 Hz MB: poly (α-cyanoacrylate n-butyl acrylate)-based MB		Sonication duration: 600 s Repeated treatment for 5 weeks		APP/PS1 mice	- ↑ local BBB opening and cognitive levels - ↑ nanoparticle release of Qc@SNPs into the brain - ↓ Aβ content and neuron loss	[135]
	Type: MRIgFUS Frequency: 1.68 MHz Acoustic pressure: Controlled by a feedback controller and allowed for consistent BBB permeabilization Burst length: 10 ms PRF: 1 Hz MB: Definity 0.02 mL/Kg		Sonication duration: 120 s single treatment (bioavailability study)/ weekly treatment for two weeks (Efficacy study)		TgCRND8 mice	- ↓ proinflammatory TNF-α - ↑ efflux of Aβ from brain - ↑ efficacy of IVIg by reducing AD pathology - ↑ neurogenesis in hippocampus	[128]

## Data Availability

Not applicable.

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
