# Peer review of "Disease-Modifying Effects of Non-Invasive Electroceuticals on β-Amyloid Plaques and Tau Tangles for Alzheimer’s Disease"

_ijms, 2022, doi:10.3390/ijms24010679_

Round 1

Reviewer 1 Report

The article entitled “Disease-modifying Effects of Non-invasive Electroceuticals on β-Amyloid Plagues and Tau Tangles for Alzheimer’s Disease” writen by Bok et al., might be of interest to th Journal’s Readers. Although this article shows numerous attempts to improve the health of Alzheimer's patients using physical means, such as ultrasound, NIR, etc., the results are still modest. Therefore, the authors replaced the less information with many explanations on various radiation types.

The manusript should check onc again for typographic mistakes and a little bit improved because it seems to be long.

Page 5, row 153: „2.1.2β-. amyloid Plaques”More correct: 2.1.2 β-Amyloid Plaques

Page 5, row 199: „Several recent studies have investigated the effects of tDCS on the pathological symptoms of AD from the cognitive behavioral level to the molecular level.” Literature?

Page 15, row 127: Is that correct? “far infrared rays (25,000 to 1,000,000 nm)”.

Author Response

Reviewer 1

The article entitled “Disease-modifying Effects of Non-invasive Electroceuticals on β-Amyloid Plagues and Tau Tangles for Alzheimer’s Disease” writen by Bok et al., might be of interest to th Journal’s Readers. Although this article shows numerous attempts to improve the health of Alzheimer's patients using physical means, such as ultrasound, NIR, etc., the results are still modest. Therefore, the authors replaced the less information with many explanations on various radiation types.

First of all, we really appreciate the reviewer’s constructive comments. In the current revised manuscript, we have addressed each issue raised by the reviewer on a point-by-point basis. Our answers to the questions are listed below, and the main text file has been modified according to the valuable comments of the referee. All revisions to the manuscript are highlighted in red.

The manuscript should check one again for typographic mistakes and a little bit improved because it seems to be long.

[Response] we are sorry for the typographic mistakes. As the reviewer pointed out, we checked out the manuscript again, and have corrected several typos in the revised manuscript. In addition, we have attempted to shorten the manuscript by removing several repetitive sentences and contents

Page 5, row 153: „2.1.2β-. amyloid Plaques”More correct: 2.1.2 β-Amyloid Plaques

[Response] As the reviewer pointed out, we have corrected it to “2.1.2 β-Amyloid Plaques” in the revised manuscript (Page 4, row 152).

Page 5, row 199: „Several recent studies have investigated the effects of tDCS on the pathological symptoms of AD from the cognitive behavioral level to the molecular level.” Literature?

[Response] As the reviewer pointed out, we have listed recent studies regarding the effect of tDCS on AD from the cognitive behavioral level to the molecular level in the revised manuscript (Page 5, row 199).

Page 15, row 127: Is that correct? “far infrared rays (25,000 to 1,000,000 nm)”.

[Response] We really appreciate your sharp comment. We have described the wavelength band by referring to the International Organization for Standardization (ISO 20473 scheme). However, as the reviewer points out, the unit representation can be misleading. Therefore, we have corrected these to (wavelength: 0.78 to 3 μm) and (50 to 1,000 μm) in the revised manuscript (Page 15, row 126)

Reviewer 2 Report

Manuscript ID: ijms-2032827 

Disease-modifying Effects of Non-invasive Electroceuticals on beta-Amyloid Plagues and Tau Tangles for Alzheimer’s Disease

This is a completely comprehensive review, covering almost all the forms of the electronic neurostimulators in the point of the reducing effects on Alzheimer pathologies including Abeta ones and tau ones. This review will be an encyclopedia of the research field of the electronic neurostimulators related with Alzheimer disease. The manuscript is well-written and almost ready for publication. The following little suggestions may help to increase further the value of this review.

(1) The list of the abbreviations used will be very helpful.

(2) In each summary table, the effect of physiological functions should be added or shown in a separated column. For example, memory or cognitive ability (increased, no change, decreased, not examined).

(3) The summary of clinical trial (present and future) may be added before "6. Conclusions and Perspectives". There seem to be only a couple of the clinical reports about the electroceuticals on Alzheimer disease. For example, transcranial magnetic stimulation is now somehow clinically applied to depression. The clinical applications of the electroceuticals to the diseases other than Alzheimer disease, the difference between these diseases and Alzheimer disease, the near future clinical prospect of the electroceuticals on Alzheimer disease and so on may be added as a clinical summary section.

(4) Line 153 "2.1.2beta-. amyloid Plaques" should be " 2.1.2 beta-amyloid Plaques"

(5) The font of the References (p26-p35) should be unified with that of the text.

(6) In Table 5, the meaning of -x (for example, "-x significantly in clinical changes) should be indicated.

End of File

Author Response

Reviewer 2

This is a completely comprehensive review, covering almost all the forms of the electronic neurostimulators in the point of the reducing effects on Alzheimer pathologies including Abeta ones and tau ones. This review will be an encyclopedia of the research field of the electronic neurostimulators related with Alzheimer disease. The manuscript is well-written and almost ready for publication. The following little suggestions may help to increase further the value of this review.

First of all, we really appreciate the reviewer’s constructive comments. In the current revised manuscript, we have addressed each issue raised by the reviewer on a point-by-point basis. Our answers to the questions are listed below, and the main text file has been modified according to the valuable comments of the referee. All revisions to the manuscript are highlighted in red.

(1) The list of the abbreviations used will be very helpful.

[Response] We totally agree with the reviewer’s point. In the revised manuscript, we have added the list of abbreviations (Page 30; Line 30~ Page 31; Line 74).

(2) In each summary table, the effect of physiological functions should be added or shown in a separated column. For example, memory or cognitive ability (increased, no change, decreased, not examined).

[Response] We appreciate the constructive comment. As the reviewer suggested, we have added the effect of physiological functions in each summary table. Since each treatment has different physiological effects, we have added the contents to the main finding without adding a new column.

(3) The summary of clinical trial (present and future) may be added before "6. Conclusions and Perspectives". There seem to be only a couple of the clinical reports about the electroceuticals on Alzheimer disease. For example, transcranial magnetic stimulation is now somehow clinically applied to depression. The clinical applications of the electroceuticals to the diseases other than Alzheimer disease, the difference between these diseases and Alzheimer disease, the near future clinical prospect of the electroceuticals on Alzheimer disease and so on may be added as a clinical summary section.

[Response] In the manuscript, we described the contents of the clinical results for each chapter. As the reviewer suggested, a summary of the clinical current status and future of electroceuticals would be helpful to improve the current manuscript. Since clinical applications can be classified according to the stimulus source, this content has been added to “Conclusions and Perspectives” in the revised manuscript (Page 30, Line 5~12).

(4) Line 153 "2.1.2beta-. amyloid Plaques" should be " 2.1.2 beta-amyloid Plaques"

[Response] we are sorry for the typographic mistakes. As the reviewer pointed out, we have corrected it to “2.1.2 β-Amyloid Plaques” in the revised manuscript (Page 4, row 152)

(5) The font of the References (p26-p35) should be unified with that of the text.

[Response] As the reviewer pointed out, we have revised the font of References.

(6) In Table 5, the meaning of -x (for example, "-x significantly in clinical changes) should be indicated.

[Response] we are sorry for the typographic mistakes. As the reviewer pointed out, we have corrected these typographic mistakes to “- no significant changes in clinical aspect” and “- no significant effect in Aβ accumulation and cognitive recognition” in Table 5.
